# A Resilience Analysis of a Medical Mask Supply Chain during the COVID-19 Pandemic: A Simulation Modeling Approach

**DOI:** 10.3390/ijerph19138045

**Published:** 2022-06-30

**Authors:** Yi Zheng, Li Liu, Victor Shi, Wenxing Huang, Jianxiu Liao

**Affiliations:** 1School of Management, Xihua University, Chengdu 610039, China; zhengyi@mail.xhu.edu.cn (Y.Z.); huangwenxing@stu.xhu.edu.cn (W.H.); 3120182301126@stu.xhu.edu.cn (J.L.); 2China Tourism Academy, Beijing 100005, China; 3Research Institute of International Economics and Management Science, Xihua University, Chengdu 610039, China; 18881997855@163.com; 4Lazaridis School of Business and Economics, Wilfrid Laurier University, Waterloo, ON N2L 3C5, Canada

**Keywords:** COVID-19, supply chain disruption, simulation modeling, supply chain resilience

## Abstract

The COVID-19 pandemic has caused severe consequences such as long-term disruptions and ripple effects on regional and global supply chains. In this paper, firstly, we design simulation models using AnyLogistix to investigate and predict the pandemic’s short-term and long-term disruptions on a medical mask supply chain. Then, the Green Field Analysis experiments are used to locate the backup facilities and optimize their inventory levels. Finally, risk analysis experiments are carried out to verify the resilience of the redesigned mask supply chain. Our major research findings include the following. First, when the pandemic spreads to the downstream of the supply chain, the duration of the downstream facilities disruption plays a critical role in the supply chain operation and performance. Second, adding backup facilities and optimizing their inventory levels are effective in responding to the pandemic. Overall, this paper provides insights for predicting the impacts of the pandemic on the medical mask supply chain. The results of this study can be used to redesign a medical mask supply chain to be more resilient and flexible.

## 1. Introduction

In the past few decades, natural disasters have frequently broken out, posing great threats to people’s lives and property losses. “The Human Cost of Disasters 2000–2019” reports that there were 4212 natural disasters worldwide from 1980 to 1999, causing approximately 1.19 million deaths, 32.5 million people affected, and about USD 1.63 trillion in economic losses [1]. Furthermore, the number of natural disasters continued to increase, with a total of 7348 incidents and the global economic losses reached approximately USD 2.97 trillion from 2000 to 2019 [1]. The year of 2020 was a devastating year for humanity, with 274 catastrophe events that cost USD 190 billion in the global economic loss [2]. From 2011 to 2018, WHO tracked 1483 epidemic events in 172 countries, such as influenza, severe acute respiratory syndrome (SARS), Middle East respiratory syndrome (MERS), Ebola, and plagues [3]. For example, the Ebola disease broke out in West Africa in 2014, with 2000 confirmed cases and more than 1000 deaths by the end of the year [4]. Compared with the considerable loss of life, the economic and social burden was even more devastating to the directly affected countries, including Guinea, Liberia, and Sierra Le one. Their farming, tourism, manufacturing and mining industries were disrupted [5,6]. It was not until 2016 that the World Health Organization announced that the Ebola epidemic had basically ended, which had caused USD 53 billion in economic and social losses [3]. The COVID-19 pandemic is a current devastating disaster to the world. It first broke out in Wuhan, China, and then spread to the whole world, with 162, 177, 376 confirmed cases and 3,364,178 confirmed deaths as of 16 May 2021. Apart from this, lots of businesses, large and small, and the global economy have been severely affected [7]. Many countries have implemented lockdown measures and travel restrictions to prevent the further spread of the epidemic [8]. As a result, industries, such as tourism, catering, transportation, entertainment, education, and training, have been severely impacted.

With lean production and globalization, supply chains are particularly vulnerable to the outbreak of natural disasters, human-made disasters, and epidemics [9]. As a result of the isolation measures in the COVID-19 pandemic, demand for many consumer goods has fallen. However, because personal protective equipment (PPE) plays a vital role in preventing the spread of the epidemic and treating patients, its demands have increased dramatically [10]. The WHO estimates that 89 million medical masks are needed every month to respond to COVID-19. For examination gloves, the number is as high as 76 million, while the international demand for goggles is 1.6 million per month. Thus, the WHO calls on industry and governments to increase manufacturing by 40% to meet the growing global demand [11]. The healthcare supply chain faces more serious problems during the pandemic. It not only suffers from supply chain disruption, but also faces a demand surge. Therefore, it is essential to supplement health emergency supplies, such as medical materials and personal protective equipment (PPE), for pandemic preparedness and prevention [12]. For the general supply chain interruption, it is mainly due to certain problems in the facilities within a certain period of time, which lead to the sluggish operation of the supply chain or temporary stagnation. The disruption of COVID-19 is not only a problem for facilities, but also causes road interruption due to the blockade, and more importantly, a chain effect, which is passed from different supply chain stages, and even seriously affects customer demand. Several scholars propose the application of backup facilities to supplement existing facilities in response to the potential effects of various natural and anthropogenic hazards can be used to reduce the risk of disruption and increase the resilience of supply chains and other logistics functions [13,14].

Given the fact that medical masks have become essential medical protective equipment, we regard medical masks as a symbol of the fight against the COVID-19 pandemic [12]. In this paper, we explore the impact of the COVID-19 pandemic on the mask supply chain by considering of ripple effect and the duration of the disruption and assess the resilience of the mask supply chain by using AnyLogistix simulation software. As a vital emergency supply, it is important to have an adequate backup supply in case of a disaster. We present the approach of simulation, which is used to locate backup facilities and adjust their service capacity in the mask supply chain. In this way, the performance and resilience of the mask supply chain are improved.

The purpose of this paper is twofold. One is to provide a simulation model which considers the duration of disruption and ripple effects to explore the short-term and long-term impacts of the COVID-19 pandemic on mask supply chain operations and performance. The other is to provide an optimization method to improve mask supply chain resilience. The proposed method in this paper is shown to be effective by conducting a risk analysis experiment. The specific steps of the mask supply chain simulation are first to establish a corresponding simulation model based on the results of the case analysis and set up corresponding scenarios for the interruption time of different facilities so as to study the performance of the supply chain caused by the different levels of interruption caused by the epidemic, and then propose to mitigate the impact of the interruption. The solution is to establish a disaster recovery center; use the center of gravity method and network optimization method to plan the location and optimize the quantity of the disaster recovery center; obtain the optimal inventory capacity of the facility; and finally, use the risk analysis section to perform improved performance evaluation and risk evaluation.

In this paper, we consider the ripple effect in the mask supply chain in China and its disruptions on the basis of dynamic control models. This paper intends to design a more reasonable and flexible supply chain of medical masks so as to prevent a shortage of masks, like that of the COVID-19 outbreak, when the next outbreak occurs suddenly. The following key questions are answered in this paper:(1)How has the COVID-19 pandemic impacted the operational and financial performance of mask supply chains?(2)How to find the location of backup facilities and optimize the mask inventory of backup facilities?(3)How to measure and improve the mask supply chain resilience and risk management?

This paper makes several key contributions. First, it supplements the research on the impact of disruptions on supply chain operations and performance through a comprehensive examination of short-term and long-term effects of disruptions in various scenarios, which consider the duration of the disruption and the propagation process. Based on earlier research focusing on the disruption effects on the supply chain [15,16], we further explore how to improve supply chain resilience by adding backup facilities to the core link of the supply chain. Second, mitigation strategies to respond to and recover from supply chain disruptions should be a priority for business managers in today’s disaster-prone environment. In this paper, we provide specific recommendations for business managers to adopt mitigation strategies from a micro perspective. Using Anylogistix, one can perform stochastic, dynamic, and comparison experiments related to facility location planning, multi-stage and multi-period SC design and planning, inventory control, transportation control, and sourcing analysis. Third, from the perspective of micro-enterprises, this paper provides guidance on how the government should deal with the interruption of the medical supplies supply chain, how to carry out contract procurement with enterprises in advance, and how to establish its own reserve center to resist risks when the epidemic comes. 

The rest of the paper is organized as follows. The literature related to this paper is reviewed in Section 2. The case study and simulation model are presented in Section 3. Section 4 details the experimental results and analysis. Section 5 investigates the use of a backup facility. Section 6 analyzes the economic benefits and resilience of the redesigned supply chain. In Section 7, we conclude the paper by outlining the main results and insights and the direction of future research.

## 2. Literature Review

In accordance with the focus of this paper, the review of the relevant literature concentrates on three relevant research streams: (i) strategies to improve supply chain resilience; (ii) the location–allocation problem for relief center/humanitarian logistics; and (iii) simulation modeling.

### 2.1. Strategies to Improve Supply Chain Resilience

Supply chain resilience has received increasing attention from scholars and practitioners due to the frequent occurrence of emergencies (e.g., natural disasters, epidemics, and accidents) that have caused supply chain disruptions [17,18,19,20]. Ponomarov and Holcomb [21] divided supply chain resilience into three parts: prepare for unexpected events, respond to disruptions, and recover from disruptions. Ribeiro and Barbosa-Povoa [9] proposed that a comprehensive supply chain resilience framework includes four elements: focus event, adaptive framing/adaptive response, speed, and performance level. In summary, resilience is the ability to reduce the losses caused by disruptions and recover from them quickly [22].

Flexibility, redundancy, visibility, and collaboration play significant roles in improving supply chain resilience [23,24,25]. In particular, the cooperation among stakeholders in the supply chain has been paid more attention to. Belhadi, Kamble, Jabbour, Gunasekaran, Ndubisi and Venkatesh [22] proposed that cooperation among stakeholders was needed to overcome the negative impact of the epidemic and accelerate the use of digital technology. Ref. [26] explored the effect of horizontal collaboration on the supply chain resilience, considering the complexity and volatility of the environment. Leat and Revoredo-Giha [27] proposed that horizontal and vertical collaborations contribute to improving the ability of the supply chain to resist risks. Lohmer et al. [28] argued that collaboration based on block chain technology should be used to improve supply chain resilience. Jüttner and Maklan [24] showed that supply chain risks and knowledge management have a positive impact on supply chain resilience.

The capacities to modify the supply chain design and planning capacity are considered the most influential technological capacities, which can enable supply chain resilience [29]. The effectiveness of advanced technologies (such as artificial intelligence and information technology) in enhancing supply chain resilience has been demonstrated by several empirical studies. Belhadi et al. [30] explored the direct and indirect effects of artificial intelligence on supply chain resilience and supply chain performance. Their findings showed that artificial intelligence has a direct impact on supply chain performance in the short term. They also suggested applying the processing capabilities of artificial intelligence to build supply chain resilience for long-term performance. Gu, et al. [31] examined the impact of two information technology patterns (exploitative versus explorative) on supply chain resilience based on the data collected from 206 manufacturers in China. Their results indicated that these two patterns complement each other in improving supply chain resilience. Advanced Industry 4.0 technologies were also considered to be beneficial in improving supply chain resilience [22,25,32,33]. Moreover, Min [34] and Lohmer, Bugert and Lasch [28] proposed blockchain technology as one of the promising technologies which enable the transparency, security and digitization of the supply chain via smart contracts and reduce supply chain vulnerability. Hence, it should be applied to supply chain resilience.

Discrete-event simulation models have been widely used to examine supply chain disruptions [16,35], and design and optimize supply chain networks considering supply uncertainty and demand uncertainty [36,37,38]. Recently, scholars have begun to use simulation modeling methods to design resilient supply chains. Ivanov and Dmitry [39] used the simulation method to design resilient supply chains capable of mitigating the ripple effect and increasing sustainability. Li, Pedrielli, Lee and Chew [18] and Lohmer, Bugert and Lasch [28] showed that information sharing and BCT-based collaboration were beneficial to improving supply chain resilience by establishing simulation models. Kaur et al. [40] developed a supply chain resilience framework and addressed the problem of the location and quantity of food distribution centers in Tier-A Cities of India by conducting Green Field Analysis experiments.

We review the literature on supply chain resilience and find that empirical research and quantitative analysis have been the most used methods in the supply chain. Simulation and optimization software integrates multiple functions such as simulation, design, and optimization. They have also been proven to be an effective tool for designing a resilient supply chain network [40,41]. Our paper enriches and extends the research on supply chain resilience by simulating the mask supply chain during the pandemic.

### 2.2. Location-Allocation Problem for Relief Center/Humanitarian Logistics

Mixed integer programming (MIP) is a popular method to address the location-allocation problem for relief centers/humanitarian logistics [42]. Exact algorithms and heuristic algorithms are commonly used to solve these MIP models. Abounacer et al. [43] proposed an exact solution approach to solve the location–transportation problem for humanitarian aid distribution centers by building a three-objective mixed integer model. With the increasing complexity of emergency network systems, the heuristic algorithm may be more suitable for solving multi-objective integer programming models [42,44]. Further, some scholars proposed improved heuristic algorithms (meta-heuristic algorithm, hybrid heuristic algorithm and math-heuristic algorithm) to solve such models more effectively [45,46,47]. Based on data modeling types and problem types, Boonmee et al. [48] divided facility location problems into four categories: deterministic, dynamic, stochastic, and robust. Their research confirmed that advanced algorithms are more efficient than exact algorithms.

Goal programming (GP) and compromise programming as well-known methodologies to solve multi-objective decision-making also have been applied to disaster recovery logistics network (DRLN) design. Bozorgi-Amiri et al. [49] proposed a compromise programming model to provide managerial insights about facility location and allocation in disaster relief systems. Fang and Li [50] and Hong and Jeong [51] combined specific location modeling goal with the DEA model to address a facility location–allocation (FLA) problem.

Other mathematical methods have also been employed to study the emergency facility location–allocation problem (LAP). Moline et al. [52] built the jurisdiction model and the travel time model to provide decision support for locating and staffing temporary disaster recovery centers (DRCs) to improve service and reduce costs. Dekle et al. [53] built a covering location model in a two-stage approach to identify the location of the disaster recovery centers. 

The facility location–allocation problem plays an important role in reducing disaster losses and improving the ability to recover from disasters. The above literature investigates this topic from a macro perspective, focusing on the location and allocation of facilities for national or regional relief centers to ensure the safety of citizens’ life and property. In contrast, our paper focuses on the location–allocation problem of the mask supply chain from a micro perspective.

### 2.3. Simulation Modeling

There are also many different approaches in articles examining supply chain disruptions. Ivanov and Dolgui [32] proposed the concept of a digital supply chain, which provides a theoretical framework for global enterprises to manage supply chain risk and resilience. Kaur, Pasricha and Kathuria [40] used AnyLogistix simulation software to find out the number of optimal distribution centers in the food supply chain of first-tier cities in India so as to improve the stability of the food supply chain during the epidemic. Taking the drug supply chain disruption in Mexico as an example, Lozano-Diez, Marmolejo-Saucedo and Aguilar [41] and Marmolejo-Saucedo et al. [54] developed the supply chain design model for a resilient supply network using AnyLogistix simulation software to reduce drug shortages in epidemic outbreaks. Prosser et al. [55] used Supply Chain Guru (Madagascar and Niger) and AnyLogistix (Guinea) modeling software to analyze and optimize the vaccine supply chain to provide decision-making guidance for improving vaccine availability in each country. Timperio et al. [38] used AnyLogistix simulation software to provide a decision-making framework for enhanced disaster preparedness in Nigeria. The above literature adopts the method of simulation modeling to design and optimize the supply chain from a macro perspective, taking a country or an industry as an example to improve the elasticity of the supply chain and enhance the ability of the supply chain to resist risks.

Ambulkar, Blackhurst and Grawe expand our understanding of factors that contribute to development of firm resilience to supply chain disruptions by using empirical examination [56]. Park, Min and Min develop a structural equation model to test causal relationships among risk taking propensity, SC security initiatives, and SC disruption occurrence. By constructing dynamic capability theory [57]. Parast examines the relationships among a firm’s R&D investment, supply chain disruption risk drivers, supply chain performance, and firm performance, using data collected from manufacturing and service organizations in the U.S [58]. Behdani and Srinivasan present an agent-based modeling framework for handling disruptions in supply chains [59]. Sarkar and Kumar investigate behavioral decision-making in multi-echelon supply chains experiencing disruptions [60].

## 3. Case Studies and Simulation Model

### 3.1. Methods

Simulation software is regarded as a suitable tool to analyze and optimize large-scale systems characterized by time dependence, randomness, and interactions. Furthermore, a dynamic simulation model can be a powerful combination of different functions with the ability to capture all operational rules and reflect all dimensions of the supply chain. In addition, for complex random problems, simulation technology can help identify the near-optimal solutions with constraints [61].

AnyLogistix (ALX) has been proven to be a successful simulation software tool for designing and optimizing. It provides various functions, including supply chain optimization, risk assessment, transportation planning, and performance visualization [32,62]. Discrete-event simulation in AnyLogistix has other advantages over other simulation software, such as its ability to accommodate random factors, such as outages, inventory, purchasing and shipment control strategies. It can reflect the change of the supply chain in detail with the change of parameters. In addition, AnyLogistix is a convenient supply chain risk assessment platform, allowing users to replicate a supply chain network and simulate its operations.

In this paper, we translate the actual problems presented in the preceding cases into corresponding mathematical models, such as formulating the facility location problem into a center of gravity analysis model and a mixed integer linear programming model and rely on the profession solver CPLEX in AnyLogistix software to solve the solution. The solutions in the paper include the severity analysis of the performance impact of node facilities under different interruption times, the planning of disaster recovery center locations, the variation experiment of inventory control, and the risk analysis of service levels.

### 3.2. Case Study

China produces about 50% of masks used in the world. China’s lockdown measures to prevent the COVID-19 pandemic have caused the disruption of mask production, leading to a severe shortage of masks [63]. It is time for the mask supply chain to develop effective measures to mitigate the risk of supply chain disruption and improve the resilience of the supply chain. To the best of our knowledge, a typical mask supply chain is primarily composed of suppliers, manufacturers, distributors, and retailers [64]. Therefore, we simulate a four-stage supply chain that includes 1 supplier (in Guangyuan), 2 factories (in Xingtai and Lishui) and 2 distribution centers (in Chengdu, Wuhan, Shenzhen, Beijing, and Shanghai), and 60 retailers (clinics) scattered all over China. The overall supply chain structure is shown in Figure 1.

The supplier transports the raw materials to two factories within three days, and these factories process the raw materials into finished products and then ship them to distribution centers. Customers across the country obtain their goods from distribution centers. Customers order 80–100 packs of masks from the distribution center every 10 days with an expected lead time between 5 and 8 days. If the delivery time of the order exceeds eight days, it is regarded as delayed delivery, which negatively affects the service level. The factories and distribution centers adopt the (s, S) inventory policy. That is, when the inventory level is below a fixed replenishment point (s), the downstream members order raw materials/products from the upstream members until the inventory quantity reaches S. To reduce cost and save time, materials and products are shipped according to the principle of nearby transport, which means that downstream supply chain members always obtain corresponding orders from nearby upstream members. Trucks with a speed of 50 km/h and a capacity of 80 m^3^ are the main transport vehicle in this supply chain network. When the supply chain network operates normally, the delivery time from the supplier to the factory is three days, the delivery time from the factory to the distribution center is two days, and the orders shipped from DCs to customers take two days. The COVID-19 pandemic broke out in Wuhan, China, and it quickly spread to the whole country and other countries around the world. The operation of the supply chain is facing enormous challenges due to facility disruptions and supply and demand shocks [8,55]. To show the propagation process of the coronavirus in details, we summarize the key events nodes from December 2019 to September 2020, as shown in Figure 2.

### 3.3. Simulation Model

We take an interest in the key factors that have a negative impact on the supply chain’s operation and performance in the COVID-19 pandemic. Thus, we examine the supply chain performance in three scenarios based on the outbreak process. The three scenarios are as follows:Factory disruption in the mask supply chain;The pandemic propagates to distribution centers;The pandemic further propagates to the market (demand increases by 10 times).

Advanced Simulation in AnyLogistix is used to simulate the different cases in the above three scenarios, considering the duration of disruption in this paper. We select these key performance indicators (KPI) to reflect the operation and performance of the supply chain, which are shown in Table 1.

In Scenario (i), we build simulation models with 30, 45, and 60 days of factory disruption and disruption-free model, respectively. In Scenario (ii), simulation models with 30, 45, and 60 days of distributer center disruption are created. We build simulation models with 30, 45, and 60 days of market disruption in Scenario (iii). The results of a total of 21 simulation experiments are shown in Table 2, Table 3 and Table 4. In our experiment, 100 replicates are created for each of these 21 simulation experiments to reduce the randomness of the output. The simulation operation lasts for one year, and the warm-up period is six months.

**Table 1 ijerph-19-08045-t001:** Key performance indicators.

Type	Indicators
**Finance**	Profit; Revenue; Total Cost; Other indicators
**Inventory**	Available Inventory; Available Inventory Backlog; On-hand Inventory
**ELT Service Level**	ELT Service Level by Orders/Products/Revenue
**Demand**	Demand (Products Backlog); Demand Placed (Products) by Customer; Demand Received (Products)
**Lead Time**	Lead Time; Max Lead Time; Mean Lead Time

The following Figure 3 shows material and information flow in the supply chain and the transport routes in normal and outbreak situations. The demand generated in the market is normal distribution, and customers order from the distribution center. The DCs/factories order products/raw materials from the upstream echelon according to the (s, S) inventory policy. When the supply chain is disrupted by the outbreak of the epidemic, the remaining capacity is randomly allocated to the downstream of the supply chain [16].

## 4. Experimental Results and Analysis

### 4.1. Performance of SC without the Epidemic

Based on the above analysis, we firstly simulate and analyze the supply chain’s operation and performance without disruption with the usage of the SIM Global Network Examination scenario in ALX, and the results are shown in Figure 4.

As shown in Figure 4a, the supply chain works well when there is no epidemic. The annual profit is USD 1,533,659.018. It is can be seen from Figure 4b that the inventory cycle time of factories and DCs in a year is consistent, which reflects the regularity and stability of the order and inventory operations. ELT service level represents the ratio of orders received on time by the customer to the total orders placed. From Figure 4c, we can see that the supply chain ELT service level is 1, indicating that there are no delayed orders. In addition, the time from the customer placing the order to receiving the goods is two days, and the total lead time is 2 × 60 = 120 days.

### 4.2. Performance of SC with the Epidemic

The supply chain has faced challenges of facility disruption and demand changes due to the outbreak of the COVID-19 pandemic [16]. We build simulation models for different scenarios which consider the duration of the disruption and the propagation progress of the epidemic to explore the impact of disruption duration and scales of epidemic propagation on supply chain operation and performance by using the random events in ALX.

#### 4.2.1. Factories Disruption

We simulate the scenarios where upstream factories are disrupted for 30, 45, and 60 days, and output key performance indicators for comparative analysis.

Table 2 shows that the supply chain revenue remains unchanged when the factories are disrupted by the epidemic, while the profit and service level drop with the disruption duration. This is because the remaining inventories in factories and distribution centers are still adequate to meet customer demand. However, factory shutdown potentially increases operation costs and transportation time. Thus, supply chain profits and service levels decline.

**Table 2 ijerph-19-08045-t002:** The supply chain performance with factory disruptions.

Scenario	Time	Profit	Revenue	Total Cost	ELT
1	0	1,533,659.018	2,328,480	794,820.982	1.000
2	30	1,444,515.296	2,328,480	883,964.704	1.000
3	45	1,439,485.154	2,328,480	888,994.846	0.956
4	60	1,415,376.715	2,328,480	913,103.284	0.912

#### 4.2.2. The Epidemic Propagates to Distribution Centers

With the further propagation of the epidemic outbreak in China, distribution centers facilities in five regions are also closed. We assume the distribution centers are disrupted for 30, 45, and 60 days, and the key performance indicators for all scenarios are summarized in Table 3.

**Table 3 ijerph-19-08045-t003:** The supply chain performance when the epidemic propagates to the DCs.

Scenario	Time	Profit	Revenue	Total Cost	ELT
5	30 30	1,254,931.776	2,134,440	879,508.224	0.917
6	30 45	1,166,057.556	2,005,080	839,022.444	0.861
7	30 60	1,120,894.358	1,940,400	819,505.642	0.833
9	45 45	1,166,057.556	2,005,080	839,022.444	0.861
10	45 60	1,120,894.358	1,940,400	819,505.642	0.833
11	60 60	1,120,894.358	1,940,400	819,505.642	0.833

From Table 3, we find that the longer the duration of the distribution center outage, the greater the negative impact on the profit and service level of the supply chain (refer to scenarios 5, 6, and 7). Customers order from the distribution center, and the long-term disruption of the distribution center potentially increases stockout. Thus, the supply chain profit and service level decline. In addition, we also find an interesting phenomenon, that is, the downstream disruption in the supply chain has a dominant impact on the operation and performance of the supply chain (refer to Scenarios 6 and 9, and Scenarios 7, 10, and 11). In other words, when the epidemic spreads downstream of the supply chain, the operation and performance of the supply chain profoundly depend on the opening and closing times of downstream facilities.

#### 4.2.3. The Epidemic Further Propagates to the Market (Demand Increases by Ten Times)

The COVID-19 pandemic has reduced commodity prices [65], while the demand for masks as medical supplies has soared [66], so we assume the demand increases by 10 times in our model. In this part, we explore the impact of facilities disruption and surge in demand on supply chain operations and profits.

It can be seen from Table 4 that when the epidemic propagates to the market and demand soars, although the supply chain sales increase due to a large number of orders, the supply chain profit drops; in particular, the service level drops sharply. Furthermore, in Scenarios 12–14, all key performance indicators remain unchanged. The reason is that when orders exceed supply chain operating capacity, the duration of market disruption has no effect on supply chain performance.

**Table 4 ijerph-19-08045-t004:** The SC performance when the epidemic propagates to market (demand increases by 10 times).

Scenario	Time	Profit	Revenue	Total Cost	ELT
12	30 30 30	1,101,177.029	4,551,480	3,450,302.971	0.128
13	30 30 45	11,011,77.029	4,551,480	3,450,302.971	0.128
14	30 30 60	1,101,177.029	4,551,480	3,450,302.971	0.128
15	30 45 45	1,001,651.986	4,176,720	3,175,068.014	0.125
16	30 45 60	1,001,651.986	4,176,720	3,175,068.014	0.125
17	30 60 60	1,103,039.443	4,143,120	3,040,080.557	0.123
18	45 45 45	1,001,651.986	4,176,720	3,175,068.014	0.125
19	45 45 60	1,001,651.986	4,176,720	3,175,068.014	0.125
20	45 60 60	1,103,039.443	4,143,120	3,040,080.557	0.123
21	60 60 60	1,103,039.443	4,143,120	3,040,080.557	0.123

In conclusion, the facility disruption and ripple effect potentially increase the risk of supply chain operations and cause economic losses. Resilience enables the supply chain to mitigate disruption and recover more quickly [28]. Thus, for supply chain members, it is necessary to adopt proactive strategies to improve supply chain resilience. By comparing the disruptions in the supply chain, it can be found that the disruptions of distribution centers have the greatest impact on the supply chain operation and performance during the propagation of the epidemic. Backup distribution centers may be of great significance in designing a resilient supply chain network [67]. Thus, in the next section, we increase the number of backup distribution center facilities, optimize their inventory, and then test the changes in supply chain resilience after the interruption.

## 5. Implementation and Results of Backup Facility

In this section, we first find out the location of these backup distribution centers through the Green Field Analysis (GFA) experiment. We use the gravity method for the facility location model, and use integer programming for the network optimization model.

In order to determine the best location for a disaster recovery center, the distance from the customer to the warehouse is calculated, determined by points weighted by the flow of product between each supplier and the potential factory, and weighted by their respective needs while meeting the minimum cost.

Network optimization considers a set of selectable locations, such as the choice of disaster recovery center in this paper (e.g., S = {GFA DC; GFA DC 2; GFA DC 3} and a set of retail stores M (60)). The set T: = SXM contains all possible logistics and transportation routes between the disaster recovery center area and the market. When any facility s ∈ S, the annual cost is incremental, and if the transport route is chosen, an additional cost is incurred. Obviously, it is useless to install a transport link between market m and facility s if s is not opened, e.g., if we set it, and at the same time, then we would end up with a useless and unrealizable solution for the network optimization. In order to avoid such a failure, we introduce the constraints that couple facility installation with transport link installation decisions and ensure that we install a transport link only if it has been decided that the origin facility should also be installed.

The buffer between supply and demand is achieved through inventory, but excessive inventory incurs costs. Therefore, finding the optimal inventory of backup facilities is critical.

### 5.1. Location of Backup Facilities

Green Field Analysis (GFA) experiment in AnyLogistix is validated for its effectiveness and extensibility [16,40]. This experiment is based on the principle of the center of gravity method to find the optimal position of the backup facilities. We obtain the locations of three backup facilities by implementing the GFA experiment. The results are shown in Table 5.

**Table 5 ijerph-19-08045-t005:** GFA DC location.

	Name	Latitude	Longitude
1	GFA DC 3	38.828	103.548
2	GFA DC 2	35.017	97.805
3	GFA DC	33.168	104.993
4	FC Xingtai	37	115
5	FC Lishui	28.46	119.91
6	DC Wuhan	30.583	114.267
7	DC Shenzhen	22.546	114.068
8	DC Shanghai	31.222	121.458
9	DC Chengdu	30.667	104.067
10	DC Beijing	39.907	116.397

### 5.2. Inventory Optimization of Backup Facilities

The variation and comparison experiment in ALX has the function of optimizing multi-level inventory with consideration of inventory holding costs (including storage costs and labor expenses), inventory turnover, inventory rates, and service level. In this case, customers place orders once every 10 days; each order quantity is 80 to 100 packs of masks; and the number of customers nationwide is 60, so the number of orders in a year is less than 100 × 3 × 12 × 60 = 216,000. Therefore, we set a few verification parameters: check inventory $\;I_{check}$ = 200,000, the service level in this inventory is defined as $ELT_{check}$. Other parameters, such as unlimited inventory, are defined as $\;I_{unlimited}$, the service level in this situation is defined as $ELT_{unlimited}$, the optimal inventory is defined as $I_{opt}$, and the optimal service level is defined as $ELT_{opt}$. The variation and comparison experiment procedure are as shown in Figure 5.

The above experiments are repeated according to the above steps, and the optimal inventory of backup facilities is obtained when the experimental step size is accurate to units of 50, as shown in the following Table 6:

## 6. Analysis of Economic Benefits and Resilience of Redesigned Mask Supply Chain

We explore the economic benefit and resilience of a redesigned supply chain by comparing the original supply chain with the redesigned one in this section.

### 6.1. Analysis of Economic Benefits

We choose Scenario 18, which is the closest to reality, as an example, and output its simulation results, as shown in Figure 6.

The simulation results show that the disruption has a negative impact on the profit and service level of the supply chain due to the increase in total cost and traffic congestion. Figure 6b shows that the inventory fluctuates evenly before the disruption occurs. During the disruption, the inventory fluctuates greatly. The factories and some DCs have a severe backlog, and some are seriously out of stock. Figure 6c indicates that when the supply chain disruptions occur, the service level drops sharply, followed by a small rebound in volatility, but the overall service level remains low. Figure 6d shows that the number of orders far exceeds the capacity of the supply chain, and orders arriving on time are at a very low level (857.65). Additionally, the lead time is prolonged.

In Section 5, we redesign the supply chain by locating backup facilities and optimizing their inventory. Here, we discuss the impact of disruption on the redesigned supply chain performance, and the simulation results are shown in the figure below.

As can be seen from Figure 7, the backup facilities alleviated the negative impact of the disruption on the supply chain operation and performance. The redesigned supply chain inventory is relatively stable facing the disruption; although the available inventory is slightly insufficient in a later period, the shortage or backlog of goods is alleviated. In addition, the early service level and order fulfillment rate are improved, and the lead time of the goods is shortened. A comparison of KPIs between the original supply chain and redesigned supply chain is shown in Table 7.

### 6.2. Risk Analysis and Resilience

In this section, we take the service level as the object to conduct the risk assessment of the supply chain. In the risk analysis experiment, we set the service level of 0.7 as the lowest point of stability (also called failure service level) and 0.95 as the recovery point. Once the service level is lower than 0.7, it means that the supply chain system has failed. If the service level is between 0.7 and 0.95, it means that the operation of the supply chain is threatened. If the service level is higher than 0.95, it means that the supply chain system returns to stability and the recovery time is calculated. The risk analysis experiment for each random event is repeated 10 times. We select some scenarios with a disruption time of 45 days for risk analysis. The experimental results are as follows Figure 8:

Figure 8a,b reflect that the supply chain has a degree of risk resistance and stability so that the supply chain can withstand the impact of short-term disruptions. However, in the face of the challenges of prolonged disruption and demand shock, the supply chain operating system may be compromised. As shown in Figure 8c, the service level is lower than 0.7 on the 126th day. The overall recovery time is 180.1 days. Figure 8d shows that the service level is lower than 0.7 on the 176th day, and the redesigned supply chain recovery time has been greatly shortened to 133.1 days.

## 7. Conclusions and Future Research

This study can support decision makers in a wide range of decisions in the context of supply prepositioning to prepare for and respond to disasters. We analyze the impact of disruption risk and the ripple effect on the design of production and distribution networks in the SC. As an example, we take a real-life case study of a severe disruption at a DC. Using simulation and optimization, we compare SC performance in the disruption-free mode and the had-disrupted SC. We also analyzed the impact of demand variability on SC performance in terms of profits, service levels, inventory and lead time.

A mask supply chain consisting of suppliers, manufacturers, distributors and retailers is simulated using AnyLogistix for potential scenarios which are most likely to occur during the spread of the COVID-19 pandemic. We explore the impact of disruptions on mask supply chain operations and performance and consider the duration of the disruption and the epidemic propagation process. The simulation results indicate that the distribution center disruption has the most significant negative impact on the mask supply chain operation and performance. Hence, backup distribution centers are added to the mask supply chain network, potential locations of these facilities are identified, and inventories are optimized by using simulation software. The major research findings are as follows.

Firstly, through the simulation of mask supply chain operation without epidemic, we find that supply chain performance and service level are excellent and can resist risks of disruption to some extent. Secondly, short-term disruptions have almost no impact on the supply chain operations and performance due to their own resilience. However, by simulating various disruption scenarios, we find that the long-term disruptions can be highly severe to the mask supply chain operation, the supply chain performance, service level, and customer satisfaction. Further, when the pandemic spreads further down the supply chain, the downstream facilities play a decisive role in supply chain performance. With the increasing duration of downstream facilities disruption, the mask supply chain performs worse. Thirdly, the Greenfield Analysis experiment is conducted to add a backup distribution center in the mask supply chain network. It identifies the potential locations and optimizes their inventory levels. Compared with the original supply chain, the total recovery time of the redesigned supply chain is shortened by 50 days, the total profit increases by USD 6,128,344.961, and the service level is improved by 45.5%. Moreover, the redesigned supply chain is more resilient: its total recovery time can be reduced by 47 days. In this paper, we suppose that all the firms in the mask supply chain share a common goal of fulfilling profits requirements. In the meanwhile, many companies have built and implemented strategies to deal with supply chain risk management and business strategies. They have built a diversified reserve center from a geographic perspective to reduce the supply-side risk from any region. Overall, this paper provides insights for improving the resilience of the medical mask supply chain and theoretical support for decision making in the medical mask supply chain.

Regarding the limitations of this paper, the simulation model is limited to a specific object, and it is difficult to accurately summarize all the details of the actual system. The first is that during the COVID-19 pandemic, the parameters required for masks are set to 10 times of the initial conditions. However, stochastic and dynamic demand is more in line with reality, especially when major emergencies break out. Second, we only considered the supply chain of one product. Third, this paper used a case study approach, which restricts the immediate generalizability of insights. To address these limitations in future research, first, one can shift attention to stochastic demand, which is more realistic. Second, one can consider multi-commodity, multi-level, and multi-period supply chain networks. Last but not least, advanced technologies, such as artificial intelligence, machine learning, and digitalization, can be incorporated into supply chain management to help reduce operational risks related to epidemic outbreaks more effectively.

## Figures and Tables

**Figure 1 ijerph-19-08045-f001:**
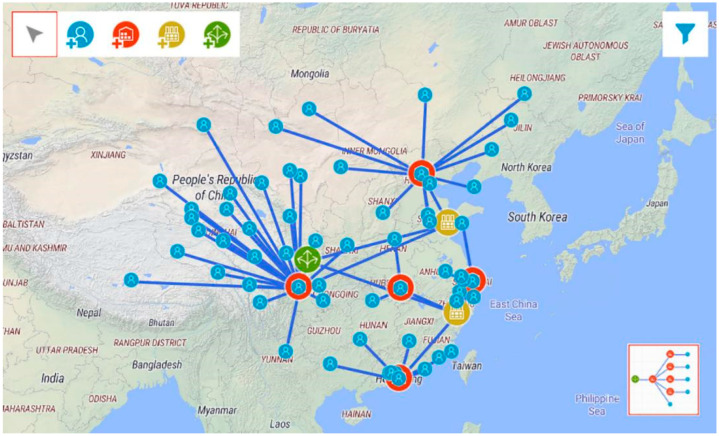
The supply chain structure. The blue icon refers to customer, the red is distribution center, the yellow is factory, and the green is supplier.

**Figure 2 ijerph-19-08045-f002:**
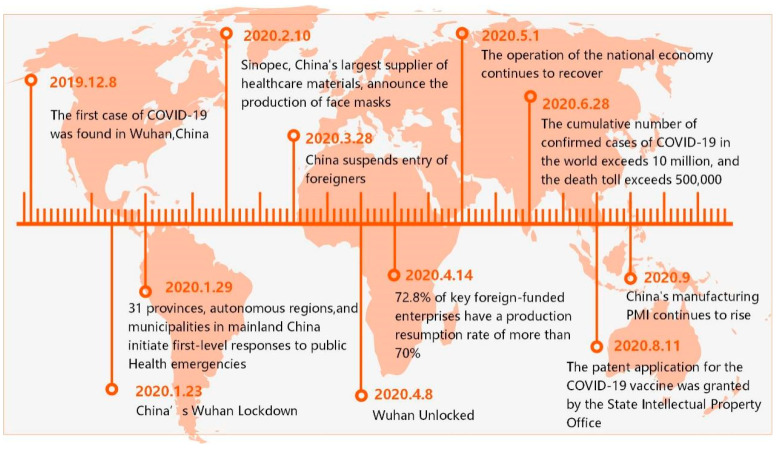
Key events of the COVID-19 pandemic.

**Figure 3 ijerph-19-08045-f003:**
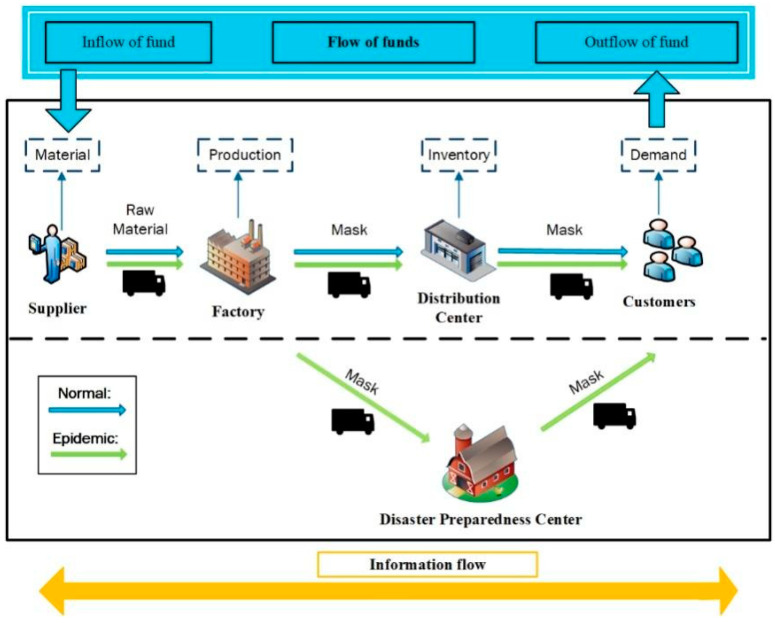
Material and information flows in supply chain.

**Figure 4 ijerph-19-08045-f004:**
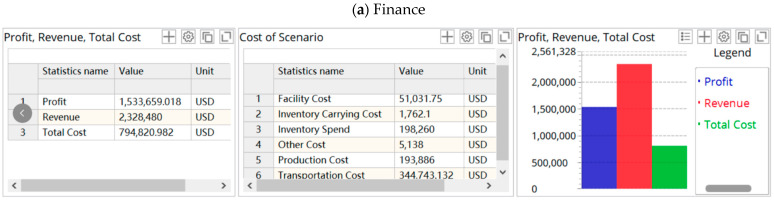
The supply chain performance in disruption-free scenario. (**a**) Finance; (**b**) inventory; (**c**) ELT service level; (**d**) demand, where the titles are Demand (Products Backlog), Demand Placed (Products) by Customer (**left**), Demand Received (Products), Fulfillment Received (Products On-time) (**middle**), Fulfillment Received (Products), Fulfillment Received (Products) by Customer (**right**); (**e**) lead time.

**Figure 5 ijerph-19-08045-f005:**
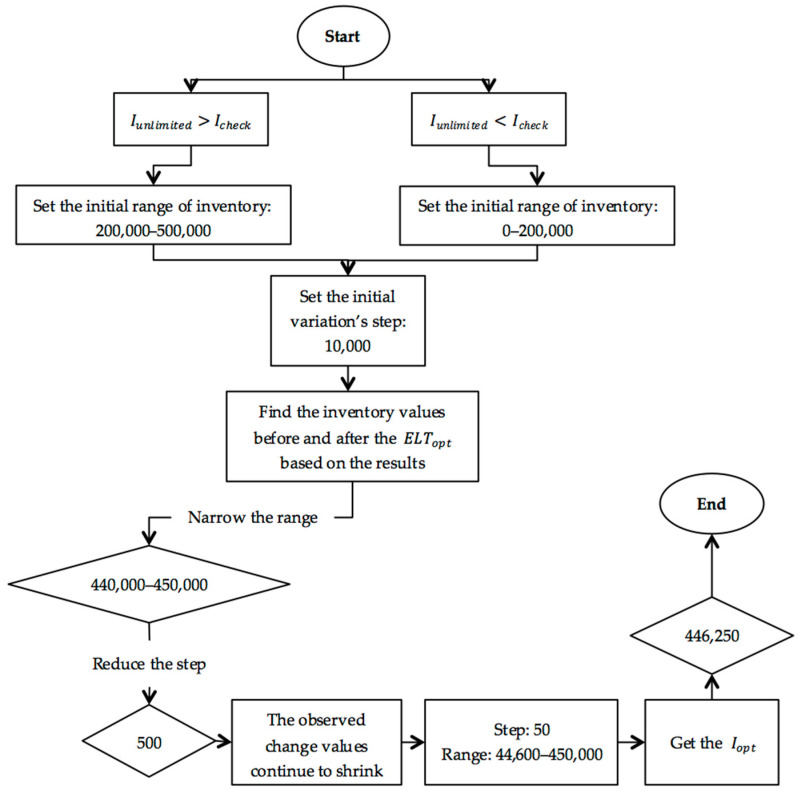
The variation and comparison experiment procedure.

**Figure 6 ijerph-19-08045-f006:**
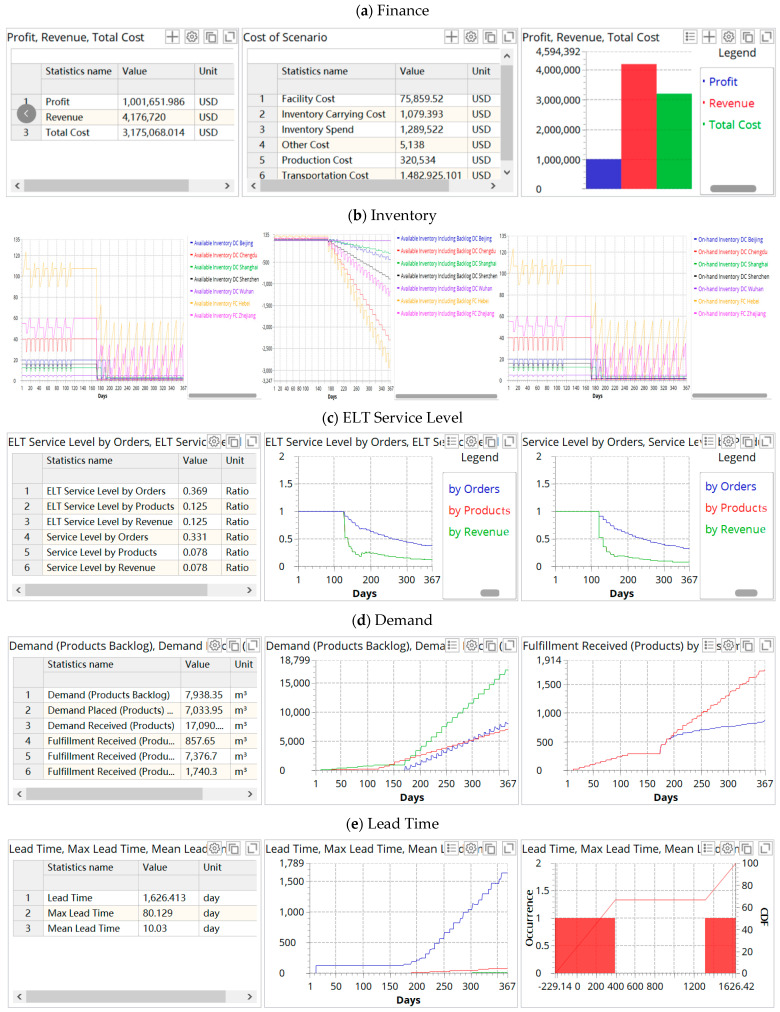
KPIs of the supply chain in Scenario 18. (**a**) Finance; (**b**) inventory; (**c**) ELT service level; (**d**) demand, where the titles are Demand (Products Backlog), Demand Placed (Products) by Customer (**left**), Demand Received (Products), Fulfillment Received (Products On—time) (**middle**), Fulfillment Received (Products), Fulfillment Received (Products) by Customer (**right**); (**e**) lead time.

**Figure 7 ijerph-19-08045-f007:**
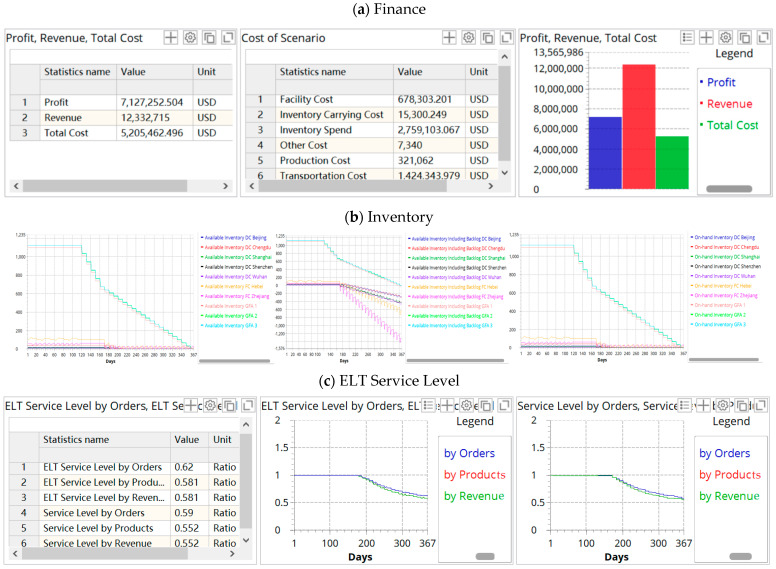
KPIs of the redesigned supply chain in scenario 18. (**a**) Finance; (**b**) inventory; (**c**) ELT service level; (**d**) demand, where the titles are Demand (Products Backlog), Demand Placed (Products) by Customer (**left**), Demand Received (Products), Fulfillment Received (Products On—time) (**middle**), Fulfillment Received (Products), Fulfillment Received (Products) by Customer (**right**); (**e**) lead time.

**Figure 8 ijerph-19-08045-f008:**
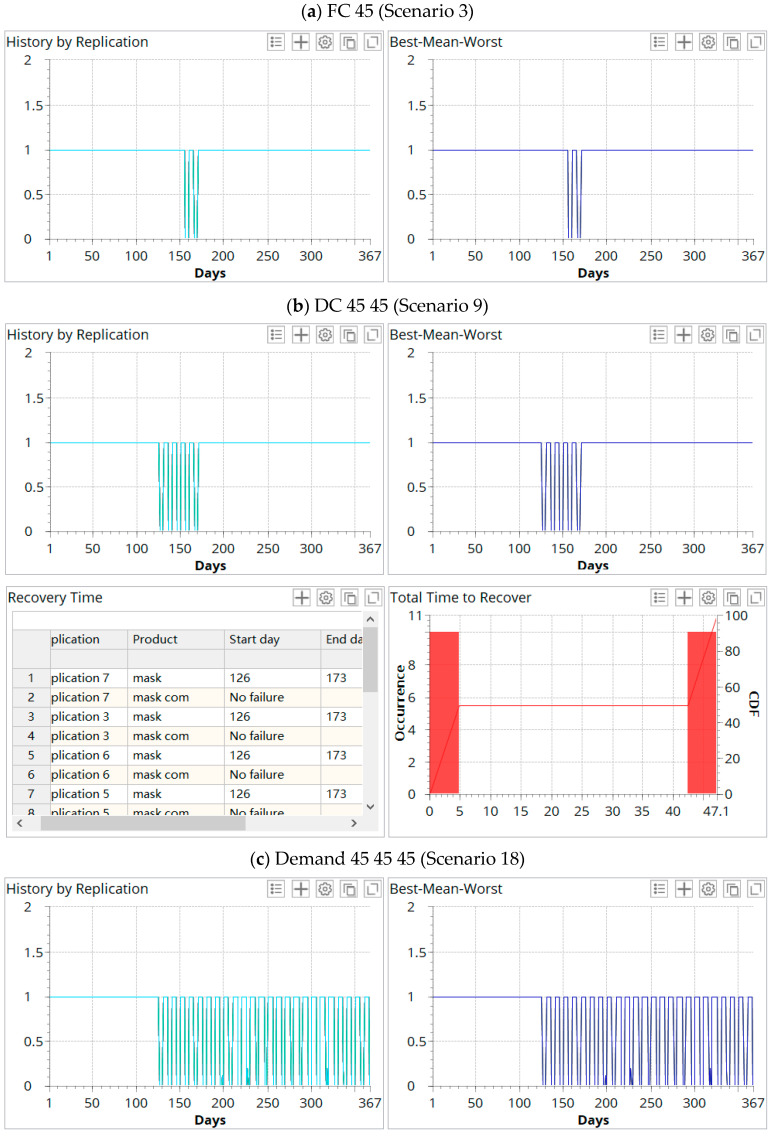
Risk analysis experiment.

**Table 6 ijerph-19-08045-t006:** The optimal inventory of backup facilities.

	1 GFA	2 GFA	3 GFA
$ELT_{check}$	0.229	0.385	0.535
$ELT_{unlimited}$	0.410	0.508	0.580
$ELT_{opt}$	0.410	0.508	0.580
$I_{opt}$	(446,250)	(281,350, 288,750)	(219,400, 224,600, 224,600)

**Table 7 ijerph-19-08045-t007:** Performance comparison.

Scenery	Profit	Revenue	Total Cost	Product Demand	Demand for Completion on Time	ELT
Scenario 18	1,001,651.986	417,6720	3,175,068.014	7033.95	857.65	0.125
Redesigned scenario 18	7,127,252.504	1.2332715	5,205,462.495	7033.95	4009.706	0.581

## Data Availability

The data used to support the findings of this study are available from the corresponding author upon request.

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
