# Peer review of "A Resilience Analysis of a Medical Mask Supply Chain during the COVID-19 Pandemic: A Simulation Modeling Approach"

_ijerph, 2022, doi:10.3390/ijerph19138045_

Round 1

Reviewer 1 Report

 In this paper, author(s) designed simulation models to investigate and predict the pandemic’s short-term and long-term disruptions on a medical mask supply chain operation using AnyLogistix, a powerful simulation and optimization software. However, I have some major comments to improve the quality of paper:

1. In the introduction section, the authors should better highlight the objectives of their work and to what extent it contributes to close the gap to the existing literature and/or practice. The innovative value of the contribution should be particularly highlighted.

2. Explain the data used in the case study in more details, the data for the testing, the accuracy.

3. The conclusion section is a bit too sparse. The authors should emphasize the impact and insights of the research and provide several solid future research directions. Also, limitations of the model should be mentioned.

4. A separate literature review section would be welcome. More recent references, published in the previous two-three years should be included.

5. Please properly improve the quality of the figures. 

6. Edit the tables properly. Also, please add more descriptions related to the tables in the paper.

7. In section 3, method section needs further clarifications.

Author Response

Responses to the Reviewers’ Comments

Dear Editors and Reviewers:

Thank you for your comments concerning our manuscript entitled “The Influence of resilience analysis of medical supplies during the COVID-19 Pandemic: A simulation modeling approach” (Manuscript Number: IJERPH-1766706). Those comments are valuable and helpful for revising and improving our paper. We have studied these comments carefully and have made changes in this revised manuscript. Please see below for our detailed responses.

Responds to Reviewer #1:

Comment (1):  In the introduction section, the authors should better highlight the objectives of their work and to what extent it contributes to close the gap to the existing literature and/or practice. The innovative value of the contribution should be particularly highlighted.

Response: Thank you for this great question. We have revised it according to your request.

Line 89-126: In this paper, we consider the ripple effect in the china mask sc and supply chain disruptions on the basis of dynamic control models.This paper intends to design a more reasonable and flexible supply chain of medical masks, so as to prevent a shortage of masks like the COVID-19 outbreak when the next outbreak occurs suddenly. The following key questions will be answered in this paper:

(1) How has the COVID-19 pandemic impacted the operational and financial performance of mask supply chains?

(2) How to find the location of backup facilities and optimize the mask inventory of backup facilities?

(3) How to measure and improve the mask supply chain resilience and risk management?

This paper makes several key contributions. First, it supplements the research on the impact of disruptions on supply chain operations and performance through a comprehensive examination of short-term and long-term effects of disruptions in various scenarios which consider the duration of the disruption and the propagation process. Based on earlier research focusing on the disruption effects on supply chain. we further explore how to improve supply chain resilience by adding backup facilities to the core link of supply chain. Second, mitigation strategies to respond to and recover from supply chain disruptions should be a priority for business managers in today’s disaster-prone environment. In this paper, we provide specific recommendations for business managers to adopt mitigation strategies from a micro perspective. Third,from the perspective of micro enterprises, this paper provides guidance on how the government should deal with the interruption of medical supplies supply chain, how to carry out contract procurement with enterprises in advance, and how to establish its own reserve center to resist risks when the epidemic comes.

 Comment (2):  Explain the data used in the case study in more details, the data for the testing, the accuracy.  

Response: Thanks for your question. The data setting of this paper is partly based on the "SIM" scenario in AnyLogistix software, which has been verified in many papers, such as the (s, S) setting strategy of inventory, random distribution of customers, etc.; the other part of the data comes from the actual supply chain scenario , such as the product price, product cost, demand, etc. involved in this article; third, the mask products involved in this article are derived from the market data investigated at the time of writing this article, including their cost, selling price, etc.

Comment (3): The conclusion section is a bit too sparse. The authors should emphasize the impact and insights of the research and provide several solid future research directions. Also, limitations of the model should be mentioned.

Response: Thank you for this great question. We have modified it in conclusions according to your request.

Line 556-565, in this paper ,we suppose that all the firms in the mask supply chain share a common goal of fufilling profits requirements. In the meanwhile, many companies have built and implemented strategies to deal with supply chain risk management and business strategies. They have build diversified reserve center from a geograghic perspective to reduce the supply side risk from any region.Overall, this paper provides insights for improving the resilience of the medical mask supply chain and theoretical support for decision-making in the medical mask supply chain.

Line 566-587, Regarding the limitations of this paper, the simulation model is limited to a specific object, and it is difficult to accurately summarize all the details of the actual system. The first is that during the COVID-19 pandemic, the parameters required for masks are set to 10 times of the initial conditions. However, stochastic and dynamic demand is more in line with reality, especially when major emergencies break out. Second, we only considered the supply chain of one product. Third, this paper used a case study approach, which restricts the immediate generalizability of insights. To address these limitations in future research, first, one can shift attention to stochastic demand which are more realistic. Second, one can consider multi-commodity, multi-level, and multi-period supply chain network. Last but not least, advanced technologies such as artificial intelligence, machine learning, and digital can be incorporated into supply chain management to help make smarter decisions to reducing operational risks related to epidemic outbreaks.

Comment (4): A separate literature review section would be welcome. More recent references, published in the previous two-three years should be included.

Response: Thank you for this great question. We have modified it according to your request.

literature4-7, 23, 26, 28, 33, 38, 47, 57-59 in the 2020 year.

Literature 8, 9, 16, 20, 21, 27, 40, 42, 46, 49, 55, 56, 61, 62 in the 2021year.

Comment (5): Please properly improve the quality of the figures. 

Response: Thank you for this great question. We have modified it according to your request.

Comment (6): Edit the tables properly. Also, please add more descriptions related to the tables in the paper.

Response: Thanks for your suggestion, we have updated the relevant table in revised manuscript.

Comment (7): In section 3, method section needs further clarifications.

Response: Thanks for your opinion. We have refined the method used in Section 3.1.

We tried our best to improve the manuscript and made some changes in the manuscript. These changes will not influence the content and framework of the paper. We appreciate for Reviewer #1s’ warm work earnestly, and hope that the correction will meet with approval. Once again, thank you very much for your comments and suggestions.

Yours,

Yi zheng

Reviewer 2 Report

The article designs simulation models to investigate and predict the pandemic’s short-term and long-term disruptions on a medical mask supply chain operation. Some aspects should be clarified, especially the scientific novelty and the methodological differences between this paper and the paper https://doi.org/10.3390/su14116529.

Is this paper only other use of the method through other application or there are novelties and differences between them?

Other comments:

References are not numbered in order of appearance in the text. Change the format.

Literature review: the papers cited are quite similar to the paper https://doi.org/10.3390/su14116529. What are the main methodological differences between this paper and the paper above?

The topics: Case studies and simulation model; and Experimental results and analysis are also quite similar too.

Table 1 of this paper and Table 3 of the above paper are also methodolocaly quite similar.

The authors didn’t cite the use of Discrete-event simulation or Mixed integer programming to optimize logistics during Covid 19 pandemic.

Author Response

Responses to the Reviewers’ Comments

Dear Editors and Reviewers:

Thank you for your comments concerning our manuscript entitled “The Influence of resilience analysis of medical supplies during the COVID-19 Pandemic: A simulation modeling approach” (Manuscript Number: IJERPH-1766706). Those comments are valuable and helpful for revising and improving our paper. We have studied these comments carefully and have made changes in this revised manuscript. Please see below for our detailed responses.

Responds to Reviewer #2:

Comment (1):  The article designs simulation models to investigate and predict the pandemic’s short-term and long-term disruptions on a medical mask supply chain operation. Some aspects should be clarified, especially the scientific novelty and the methodological differences between this paper and the paper https://doi.org/10.3390/su14116529. 

Response: Thank you for your suggestion. The novelty of the previous paper mainly considered the flexibility and robustness of the retail industry in responding to emergencies in the context of new retail. The specific method is to use simulation to simulate the ability of retail stores to withstand interruption events in the scenario of a wireless platform. This paper first examines the impact of different levels of disruption caused by the epidemic on the performance of the supply chain and provides some management insights. On this basis, the solution measures to alleviate the impact of interruption are proposed, namely establishing a disaster recovery center, including site selection and quantity optimization based on the center of gravity method, and site selection and inventory optimization based on mutation experiments. Finally, use the risk analysis section to conduct improved performance evaluation and risk evaluation. In general, this article is more complete in structure and logic, and the analysis methods used are more complex and comprehensive. It is a further application of AnyLogistix software functions and an in-depth analysis of the impact of the epidemic.

Comment (2):  Is this paper only other use of the method through other application or there are novelties and differences between them? 

Response: Thanks for your question. First of all, some of the principles we use are similar. The initial research on the impact of the epidemic was run through a simulation model, but on this basis, there are also large differences and innovations, such as the center of gravity method site selection and mutation-based experiments studied in this paper. The risk analysis, risk analysis and recovery, interrupt time, etc., are the in-depth research and innovation parts of this paper.

Comment (3):  References are not numbered in order of appearance in the text. Change the format.

Response: Thank you for this great question. We have modified it in conclusions according to your request.

Comment (4): Literature review: the papers cited are quite similar to the paper https://doi.org/10.3390/su14116529. What are the main methodological differences between this paper and the paper above? 

Response: Thanks for your question. First of all, the analysis in this paper is based on the same simulation software, and it is bound to have a certain similarity. In this paper, the centroid method location analysis, mutation experiment, and risk analysis experiment involved are not involved in the previous paper, so there are already big differences in methods.

Comment (5): The topics: Case studies and simulation model; and Experimental results and analysis are also quite similar too.

Response: Thanks for your question. Both articles are based on the background of the epidemic and the analysis of simulation software as a tool, so the experimental analysis has certain similarities. However, the focus of the two is completely different: the paper you mentioned mainly considers the flexibility and robustness of the retail industry in responding to emergencies in the context of new retail, while this paper provides measures for the medical industry to respond to public health and plans and provide inspiration for the medical industry.

Comment (6): Table 1 of this paper and Table 3 of the above paper are also methodolocaly quite similar.

Response: Thanks for your question! First of all, both articles are based on the background of the epidemic, and the starting point is to study the impact of the epidemic on different industries. Secondly, in analyzing the impact of the epidemic, the AnyLogistix software is used for analysis, so there is a certain similarity in the setting of parameters.

Comment (7): The authors didn’t cite the use of Discrete-event simulation or Mixed integer programming to optimize logistics during Covid 19 pandemic.

Response: Thanks for your question. In this paper, we mainly study the impact of the epidemic on the medical supply chain industry and solutions, and finally provide relevant management opinions and suggestions. For the logistics during the epidemic period, the logistics network optimization model (NO) based on integer programming is used in the transportation problem after site selection, such as the objective function of minimum cost and the 0-1 constraint of whether the facility is open or not, and at the same time It is also a way to verify whether the site selection is valid. It's just that in the software operation, the operation process is relatively simple, so there is not much elaboration. In addition, under the question of review 4, we also show the relevant modeling process and formula of the integer programming optimization of the NO model.

We tried our best to improve the manuscript and made some changes in the manuscript. These changes will not influence the content and framework of the paper. We appreciate for Reviewer #2s’ warm work earnestly, and hope that the correction will meet with approval. Once again, thank you very much for your comments and suggestions.

Yours,

Yi zheng

Reviewer 3 Report

The authors presented a very interesting and up-to-date research approach in which they design simulation models to investigate and predict the pandemic's short-term and long-term disruptions on a medical mask supply chain operation. The topic is interesting and noteworthy, due to the impact that the pandemic has had on global supply chains, which are still struggling with many of its effects. Despite the overall positive evaluation of the manuscript, it needs to be improved in the following areas:

1/ The research gap should be better described in the introduction. The authors mention in this context only 2 publications by one author: Ivanov (2020, 2021). What about other authors who have dealt with this issue? Further, the authors write that "In this paper, we provide specific recommendations for business managers to adopt mitigation strategies from a micro perspective" (lines 99-100): then is the micro perspective something that distinguishes this article from other publications? If so, I would like to know which publications present a different perspective...

2/ In the conclusions section, I would like to read the theoretical contribution of this manuscript to science (theory). In addition, there is no information about any of the practical and social implications of the manuscript that the authors announced in the introduction. Complementing the manuscript in this area is necessary taking into account the practical nature of the manuscript.

3 / Please correct minor linguistic mistakes, e.g. line 44: Instead of 'Coivd' should be 'Covid'

Author Response

Responses to the Reviewers’ Comments

Dear Editors and Reviewers:

Thank you for your comments concerning our manuscript entitled “The Influence of resilience analysis of medical supplies during the COVID-19 Pandemic: A simulation modeling approach” (Manuscript Number: IJERPH-1766706). Those comments are valuable and helpful for revising and improving our paper. We have studied these comments carefully and have made changes in this revised manuscript. Please see below for our detailed responses.

Responds to Reviewer #3:

Comment (1):  The research gap should be better described in the introduction. The authors mention in this context only 2 publications by one author: Ivanov (2020, 2021). What about other authors who have dealt with this issue? Further, the authors write that "In this paper, we provide specific recommendations for business managers to adopt mitigation strategies from a micro perspective" (lines 99-100): then is the micro perspective something that distinguishes this article from other publications? If so, I would like to know which publications present a different perspective.... 

Response: Thanks for your question. Ivanov’s articles analyze the structure of supply chain from a micro perspective,using anylogostics, one can perform stochatic, dynamic, and comparison experiments related to facility location planning, multi-stage,and muliti-period SC design and planning ,inventory control, transportation control, and sourcing analysis.

Comment (2):  In the conclusions section, I would like to read the theoretical contribution of this manuscript to science (theory). In addition, there is no information about any of the practical and social implications of the manuscript that the authors announced in the introduction. Complementing the manuscript in this area is necessary taking into account the practical nature of the manuscript.

Response: Thanks for your question. Thank you for this valuable suggestion. We have revised the conclusion section of the manuscript.

Line:548-555:This study can support decision-makers in a wide range of decisions in the context of supply prepositioning to prepare for and respond to disasters. we analysed the impact of disruption risk and the ripple effect on the design of production and distribution networks in the SC. As an example, we took a real-life case-study of a severe disruption at a DC. Using simulation and optimisation, we compared SC performance in the disruption-free mode and the had disrupted SC . We also analysed the impact of demand variability on SC performance in terms of profits, service levels, inventory and lead time.

Comment (3):  Please correct minor linguistic mistakes, e.g. line 44: Instead of 'Coivd' should be 'Covid'

Response: Thank you! We have corrected 'Coivd' into 'Covid' in revised manuscript.

We tried our best to improve the manuscript and made some changes in the manuscript. These changes will not influence the content and framework of the paper. We appreciate for Reviewer #3s’ warm work earnestly, and hope that the correction will meet with approval. Once again, thank you very much for your comments and suggestions.

Yours,

Yi zheng

Reviewer 4 Report

The paper investigates the impact of supply chain disruptions related to the COVID-19 pandemic on the operational and financial performances of a medical mask supply chain. The paper employs a simulation modeling approach to investigate the supply chain operational performance under different disruption and mitigation scenarios. Below are my comments to the authors:

Title

·        I would like to suggest the authors to revise the title by changing the sentence “A simulation experiment” to “a simulation modeling approach”.

Abstract

·        In the abstract, the research gap is not articulated.

·        Furthermore, the research objective and the adopted research methodology should be put in context. The authors should focus on the research methodology in general and not on the simulation software which is one of the tools employed within the methodology.

·        The key contribution should be emphasized. The contribution could be the key conclusion that can be beneficial to relevant supply chain practitioners and managers.

Keywords

·        Simulation modeling is an important keyword that should be included in this set of keywords and other important keywords may be added without exceeding the journal’s keywords limits.

1. Introduction

·        I suggest merging and summarize the first two paragraphs of the introduction section.

·        The motivation of this research should be clarified further in the introduction. The research gap should be articulated within the context of the previous related research to supply chain disruption in general, and COVID-19 disruption.

·        The research objectives and the adopted research methodology should be clarified in an organized manner.

2. Literature Review

·        I suggest the authors to include the modeling methodology as a third research stream and survey the related papers adequately. It is important to discuss the different modeling approaches (analytical modeling, simulation, etc.) that have been adopted in the literature to investigate supply chain disruptions.

3. Case studies and simulation model

·        I suggest the authors to start with outlining the modeling methodology followed in this paper to model analyze the supply chain under disruptions. Then, the case study is presented.

·        At line 227, adjust the word “blew” into “below.”

·        Careful English language editing is required for the paragraph starting from 251 to 259.

·         The authors should provide the state variables and the mathematical expressions that govern the supply chain operation. Any modeling assumptions used to develop the simulation model should be provided.

·        Further clarifications for the key performance indicators in Table 1 should be provided. The related mathematical expressions that indicate how those performance measures are calculated should be provided. Also, any related references to those performance measures should be cited.

4. Experimental results and analysis

·        The simulation model validation results should be presented and discussed before presenting the main simulation experiments.

·        At line 291, please provide further clarification for “SIM Global Network Examination.”

·        All the main results should be presented in clear tables and figures (not as screenshots from the simulation program outputs).

·        The design of the conducted simulation experiments should be clarified, and the related results should be discussed thoroughly.

·        Describe the demand models that have been considered in the different scenarios.

·        At line 323, clarify how the factory’s revenue stays unchanged at the different disruption levels.

5. Implementation and results of backup facility

·        At line 377, clarify how the number of backup facilities are determined to be 3 facilities.

·        Please improve the quality of the flowchart in Figure 6, indicate the logic and input/output of this flowchart, and provide a clear description for it.

·        Have the authors investigated other mitigation strategies like increasing inventory levels, increasing the capacity of the existing facilities, etc.

6. Analysis of economic benefits and resilience of redesigned mask supply chain

·        Provide further details regarding the redesign of the supply chain.

·        The design of simulation experiments related to this section should be described clearly.

·        The simulation results in this section should be presented and discussed clearly.

·        The statistical analysis of the results is not provided.

7. Conclusions

·        Clear managerial implications should be provided.

General Comments

·        The quality of all figures in this paper should be improved.

Author Response

Responses to the Reviewers’ Comments

Dear Editors and Reviewers:

Thank you for your comments concerning our manuscript entitled “The Influence of resilience analysis of medical supplies during the COVID-19 Pandemic: A simulation modeling approach” (Manuscript Number: IJERPH-1766706). Those comments are valuable and helpful for revising and improving our paper. We have studied these comments carefully and have made changes in this revised manuscript. Please see below for our detailed responses.

Responds to Reviewer #4:

Comment (1):  

Title· I would like to suggest the authors to revise the title by changing the sentence “A simulation experiment” to “a simulation modeling approach”. 

Response: Thank you! We have modified “A simulation experiment”to“A simulation modeling approach”in revised manuscript.

Comment (2):  

 Abstract·In the abstract, the research gap is not articulated.

 Response: Thanks for your question. We have modified it according to your request.

Comment (3): Furthermore, the research objective and the adopted research methodology should be put in context. The authors should focus on the research methodology in general and not on the simulation software which is one of the tools employed within the methodology. 

Response: Thanks for your valuable advice! We did spend too much space on the emphasis on software and neglected the method, which was our mistake. Under your suggestion, we also emphasized the specific method in the original text, and demonstrated the relevant principles.We have modified it.

Comment (4): The key contribution should be emphasized. The contribution could be the key conclusion that can be beneficial to relevant supply chain practitioners and managers.

Response: Thank you for this valuable suggestion. We have modified it according to your request.

Comment (5):

Keywords · Simulation modeling is an important keyword that should be included in this set of keywords and other important keywords may be added without exceeding the journal’s keywords limits.

Response: Thank you for this valuable suggestion. We have added “Simulation modeling” in keywords.

Comment (6):

  1. Introduction

I suggest merging and summarize the first two paragraphs of the introduction section.

Response: Thank you for this valuable suggestion. We have modified it according to your request.

Comment (7): The motivation of this research should be clarified further in the introduction. The research gap should be articulated within the context of the previous related research to supply chain disruption in general, and COVID-19 disruption.

Response: Thank you for your suggestion. We have completed the introduction section. For the general supply chain interruption, it is mainly due to certain problems in the facilities within a certain period of time, which lead to sluggish operation of the supply chain or temporary stagnation. And the disruption of COVID-19 is not only a problem with facilities, but also road interruption due to the blockade, and more importantly, a chain effect, which is passed from different supply chain stages, and even seriously affects customer demand.

Comment (8):  The research objectives and the adopted research methodology should be clarified in an organized manner.

Response: Thanks for your question. The research objectives of this paper are two-fold. One is to explore the short- and long-term impact of the COVID-19 outbreak on the disruption and chain reaction of mask supply chains in medical facilities. The second is to consider the feasibility of establishing a disaster recovery center to optimize the flexibility of the medical mask supply chain. The research methods generally include literature indexing method, case analysis method and system simulation method. The specific steps of the system simulation are to first establish a corresponding simulation model based on the results of the case analysis, and set up corresponding scenarios for the interruption time of different facilities, so as to study the performance of the supply chain caused by the different levels of interruption caused by the epidemic, and then propose to mitigate the impact of interruption. The solution is to establish a disaster recovery center; then use the center of gravity method and network optimization method to plan the location and optimize the quantity of the disaster recovery center; Obtain the optimal inventory capacity of the facility; finally, use the risk analysis section to perform improved performance evaluation and risk evaluation.

Comment (9):

  1. Literature Review

 I suggest the authors to include the modeling methodology as a third research stream and survey the related papers adequately. It is important to discuss the different modeling approaches (analytical modeling, simulation, etc.) that have been adopted in the literature to investigate supply chain disruptions.

Response: Thank you for this valuable suggestion. Thank you for this valuable suggestion. We have added section 2.3 of the literature to the revised manuscript.

Comment (10): 

  1. Case studies and simulation model

 I suggest the authors to start with outlining the modeling methodology followed in this paper to model analyze the supply chain under disruptions. Then, the case study is presented.

Response: Thank you for this valuable suggestion. We have modified it according to your request.

 Comment (11):   At line 227, adjust the word “blew” into “below.”

Response: Thank you! We have modified the word “blew” into “below” in revised manuscript.

Comment (12):  Careful English language editing is required for the paragraph starting from 251 to 259.

Response: Thank you for your suggestion, we have carefully checked and revised the English language in the revised manuscript.

  Comment (13): The authors should provide the state variables and the mathematical expressions that govern the supply chain operation. Any modeling assumptions used to develop the simulation model should be provided. 

Response:Thanks for your question! We have shown the principles and algorithms of the relevant models as follows, which are also described in detail in the software manual.

  1. Facility location model based on the center of gravity method:
  • Model assumptions
  • Transportation costs are linearly proportional to distance and volume of transportation (i.e. demand)
  • The expected transportation cost from the disaster recovery center to the customer site is equal to the product of distance and demand.
  • Parameter design

The location of the disaster recovery center

customer's location

Customer demand for our products

Distance from disaster recovery center to customers

  • Modeling

In order to determine the best location for a disaster recovery center, the distance from the customer to the warehouse is calculated, determined by points weighted by the flow of product between each supplier and the potential factory, and weighted by their respective needs while meeting the cost minimum. In this regard, we use the center of gravity method to construct the model, taking the decision variable and profit minimization as the objective function, the optimal point determined by calculus, the determined first-order derivative and zero, respectively, to obtain expressions (4), (5) to indicate the specific location of the disaster recovery center.

Based on the above principle of location selection and applying it to the GFA greenfield analysis simulation implementation in anyLogistix, we select and accurately locate the disaster recovery center.

  1. A network optimization model based on integer programming:

Network optimization considers a set of selectable locations, such as the choice of disaster recovery center in this paper (eg S = {GFA DC; GFA DC 2; GFA DC 3} and a set of retail stores M (60)). The set T:=SXM contains all possible logistics and transportation routes between the disaster recovery center area and the market. When any facility s ∈ S, the annual incremental cost is , and if the transport route is chosen, an additional cost is incurred. Therefore, the simplified form of the mixed integer programming model used is as follows:

Objective function:

Binary decision variable Indicates the open decision of the disaster recovery center, belonging to 0-1 variable. is a decision on whether to use the transportation route T between the disaster recovery center and the retail store, and if each retail store must be provided with masks by one disaster recovery center, the constraints must be adhered to:

Minimize the sum of the total annual costs by changing the above decision variables。Obviously, it is useless to install a transport link between market m and facility s if s is not opened, e.g., if we set  if, and at the same time,  then we would end up with a useless and unrealizable solution for the network optimization. In order to avoid such a failure, we introduce the constraints that couple facility installation with transport link installation decisions and ensure that we install a transport link only if it has been decided that the origin facility should also be installed.

Comment (14): Further clarifications for the key performance indicators in Table 1 should be provided. The related mathematical expressions that indicate how those performance measures are calculated should be provided. Also, any related references to those performance measures should be cited.

Response: The key indicators in Table 1 are derived from the output of the simulation results, but are not limited to the listed key performance indicators. We mainly choose financial indicators, inventory indicators, service level indicators, demand indicators and lead time represented by cost, profit, and income as the analysis objects! Other indicators are also relevant, but we have selected the most meaningful performance indicators that we think are suitable for this analysis.

Comment (15): 

  1. Experimental results and analysis

The simulation model validation results should be presented and discussed before presenting the main simulation experiments.

Response: Thanks for your question. First, before conducting the formal study, we conducted 100 repeated experiments and periodic warm-ups to ensure the accuracy and validity of the simulation model, which are also mentioned in Section 3.3. At the same time, this model is set based on the SIM basic model, that is, a complete supply chain model can be constructed only when the relevant required parameters are reached, so it is proved reasonable by the software. At the same time, most of the functions of the software model have been verified and applied by Ivanov and other authors.

Comment (16): At line 291, please provide further clarification for “SIM Global Network Examination.” 

Response: Thanks for your question. “SIM Global Network Examination.” is a case of a global supply chain that has been verified by large-scale personnel. And simulation experiment is used to model the actual products delivery on the GIS map with detailed statistics, which show the data collected on different types of facilities involved in the supply chain scenario during the experiment: Profit and lost statement, Total factory shipped, Peak capacity, Service Level.

Comment (17): All the main results should be presented in clear tables and figures (not as screenshots from the simulation program outputs).

Response: Thanks for your valuable advice. We have revised the figures and tables of the revised manuscript.

Comment (18): The design of the conducted simulation experiments should be clarified, and the related results should be discussed thoroughly.

Response: Thank you for your suggestion. The fourth part mainly studies the impact of different supply chain nodes and changes in demand brought about by COVID-19 on supply chain performance. It mainly includes the analysis of FC interruption, the propagation of FC interruption to DC interruption, and the performance impact under the influence of interruption, so as to obtain management insights such as knock-on effects and risk resistance.

Comment (19): Describe the demand models that have been considered in the different scenarios.

Response: Thanks for your question. This paper studies three scenarios (1) Factory disruption in the mask supply chain; (2) The pandemic propagates to distribution centers; (3) The pandemic further propagates to the market (demand increases by 10 times). In the first two scenarios, the impact of the epidemic has not yet spread to customers or retailers, so the demand considered is still the market demand during the research period. In the third scenario, because the outbreak of the epidemic has attracted people's attention, the demand has expanded 10 times.

 Comment (20): At line 323, clarify how the factory’s revenue stays unchanged at the different disruption levels.

Response: Thanks for your question. The above situation only appeared in the scene of FC interruption. As a direct production facility for masks, the factory played a huge role during the epidemic, and there was relatively sufficient inventory to meet the needs of various places. Therefore, if the interruption time is not long, the existing inventory of the factory can fully cope with the negative impact of the interruption, thereby ensuring the level of income.

 Comment (21):

  1. Implementation and results of backup facility

 At line 377, clarify how the number of backup facilities are determined to be 3 facilities.

Response: Thanks for your question. First of all, the software we use is the student version, so there is a certain limitation in the number of facility settings, 3 is the maximum number of facilities we can achieve. Secondly, in the process of research, a function is also made for the relationship between the number of added facilities and performance. When the number of facilities is 5-6, it is the optimal number of facilities. Too many facilities will bring greater cost. In the end, we only select the appropriate number of facilities for analysis within the feasible range. The research conducted by taking 3 facilities as an example obtains a certain research law rather than an absolute number, so we think it is reasonable.

Comment (22): Please improve the quality of the flowchart in Figure 6, indicate the logic and input/output of this flowchart, and provide a clear description for it.

Response: Thank you for your suggestion. We have made changes in the original text!

Comment (23): Have the authors investigated other mitigation strategies like increasing inventory levels, increasing the capacity of the existing facilities, etc.

Response: Thanks for your question. We have investigated the proposed mitigation strategies in this article. In fact, the essence of the disaster recovery center is the distribution center. In theory, adding a disaster recovery center is to increase the facility capacity of the distributor to alleviate interruptions and demand surges. burden on retailers. In addition, this paper finds out the optimal inventory capacity of the disaster recovery center through the mutation experiment, which keeps the inventory level at the highest theoretical level considering the management cost.

Comment (24):

  1. Analysis of economic benefits and resilience of redesigned mask supply chain

Provide further details regarding the redesign of the supply chain.

Response: Thanks for your question. The supply chain redesign is a response to the dramatic impact of the pandemic. It mainly selects scenario 18 as an example for analysis, simulates the appropriate number of disaster recovery centers through the principle of the center of gravity method, and calculates the optimal inventory quantity of each disaster recovery center through mutation experiments. That is, the 3 disaster recovery centers after the location selection and the matching inventory quantity are added to the scene 18, and other parameters are not changed to ensure the unity of variables.

Comment (25): The design of simulation experiments related to this section should be described clearly.

Response: Thank you for your suggestion. We describe it further in the article.

Comment (26): The simulation results in this section should be presented and discussed clearly.

 Response: Thanks for your question. The sixth part is mainly about the simulation analysis and performance output of the scenario after the establishment of the disaster recovery center, which is described in detail in the original text, and the other part is the risk analysis of the improved supply chain to evaluate the robustness of the improved supply chain.

Comment (27): The statistical analysis of the results is not provided.

 Response: Thanks for your question. In the 6 chapter, through the comparison of the performance output before and after the improvement, the comparison and analysis are made with various aspects such as revenue, cost, inventory, etc., to achieve the purpose of improving the performance of the supply chain and enhancing the ability of the supply chain to resist risks. Based on your comments, we have rectified the shortcomings in the original text.

Comment (27):

  1. Conclusions

 Clear managerial implications should be provided.

Response: Thank you for this valuable suggestion. We have added managerial implications in the revised manuscript.

Comment (28):

General Comments: The quality of all figures in this paper should be improved.

Response: Thanks for your question! This is an oversight on our part, we have made changes and shown in the revised manuscript.

We tried our best to improve the manuscript and made some changes in the manuscript. These changes will not influence the content and framework of the paper. We appreciate for Reviewer #4s’ warm work earnestly, and hope that the correction will meet with approval. Once again, thank you very much for your comments and suggestions.

Yours,

Yi zheng

Round 2

Reviewer 1 Report

The paper has been significantly improved in the revised version. I have no further comment. 

Author Response

Thank you for your comments concerning our manuscript entitled “The Influence of resilience analysis of medical supplies during the COVID-19 Pandemic: A simulation modeling approach” (Manuscript Number: IJERPH-1766706). Those comments are valuable and helpful for revising and improving our paper.

Reviewer 4 Report

The authors have replied satisfactorily to all the concerns I had on the earlier version of this manuscript. However, there are some further comments that should be addressed before publication.  Below are those comments:

The authors should extend Section 2.3 on the simulation modeling approach to consider a wider scope on the subject, not only focusing on the related research that adopted AnyLogistix.

Please insert the facility location model and all other mathematical models indicated in the review report in the proper places in the manuscript. Please provide the proper reference and descriptions for these models.

The clarity of some figures is missing. Please review all figures and improve their quality, clarity, and readability. For example, the legends of some figures are hard to read.

An extensive proofreading is required.

Author Response

Responses to the Reviewers’ Comments

Dear Editors and Reviewers:

Thank you for your comments concerning our manuscript entitled “The Influence of resilience analysis of medical supplies during the COVID-19 Pandemic: A simulation modeling approach” (Manuscript Number: IJERPH-1766706). Those comments are valuable and helpful for revising and improving our paper. We have studied these comments carefully and have made changes in this revised manuscript. Please see below for our detailed responses.

Responds to Reviewer #4:

Comment (1):  

The authors should extend Section 2.3 on the simulation modeling approach to consider a wider scope on the subject, not only focusing on the related research that adopted AnyLogistix.

Response: Thanks for your valuable advice! We add the literature on related methods of supply chain disruption.

There are also many different approaches in articles examining supply chain disruptions, Ambulkar, Blackhurst and Grawe expands our understanding of factors that contribute to development of firm resilience to supply chain disruptions by using empirical examination[1]. Park, Min and Min develop a structural equation model to test causal relationships among risk taking propensity, SC security initiatives, and SC disruption occurrence. By constructing dynamic capability theory[2]. Parast examines the relationships among a firm's R&D investment, supply chain disruption risk drivers, supply chain performance, and firm performance, using data collected from manufacturing and service organizations in the U.S[3]. Behdani and Srinivasanpresents an agent-based modelling framework for handling disruptions in supply chains[4] . Sarkar and Kumar investigate behavioral decision-making in multi-echelon supply chains experiencing disruptions[5].

[1]Ambulkar, S., Blackhurst, J. and Grawe, S. Firm's resilience to supply chain disruptions: Scale development and empirical examination. Journal of Operations Management, 33-34: 111-122. https://doi.org/10.1016/j.jom.2014.11.002

[2]Kihyun Park, Hokey Min, Soonhong Min,Inter-relationship among risk taking propensity, supply chain security practices, and supply chain disruption occurrence,Journal of Purchasing and Supply Management,Volume 22, Issue 2,2016,120-130.

[3]Mahour Mellat Parast,The impact of R&D investment on mitigating supply chain disruptions: Empirical evidence from U.S. firms,International Journal of Production Economics,Volume 227,2020,107671,ISSN 0925-5273,

[4]Behzad Behdani, Rajagopalan Srinivasan, Managing supply chain disruptions: an integrated agent-oriented approach,Editor(s): Antonio Espuña, Moisès Graells, Luis Puigjaner, Computer Aided Chemical Engineering, Elsevier, Volume 40,2017,595-600,

[5]Sourish Sarkar, Sanjay Kumar,A behavioral experiment on inventory management with supply chain disruption,International Journal of Production Economics,Volume 169,2015,169-178.

Comment (2):  

Please insert the facility location model and all other mathematical models indicated in the review report in the proper places in the manuscript. Please provide the proper reference and descriptions for these models.

Response: Thanks for your valuable advice! In the fifth chapter of the paper, a logical discussion of mathematical modeling methods has been added. The core of this article is to use simulation method to study the problem. Mathematical models supporting anyLogistix simulation are relatively simple, and it is more clear and understandable to demonstrate with experimental methods, such as GFA. For example, in the paper (1-4) that they did not show the mathematical model supporting the simulation process in the body, and they used experimental methods for analysis.

  1. Kaur, G.; Pasricha, S.; Kathuria, G. Resilience Role of Distribution Centers amid COVID-19 Crisis in Tier-A Cities of India: A Green Field Analysis Experiment. Journal of Operations and Strategic Planning 2020, 3, 226-239, doi:https://doi.org/10.1177/2516600X20970352.
  2. Ivanov D . Predicting the impacts of epidemic outbreaks on global supply chains: A simulation-based analysis on the coronavirus outbreak (COVID-19/SARS-CoV-2) case[J]. Transportation Research Part E: Logistics and Transportation Review, 136.
  3. Ivanov, D. Simulation-based single vs. dual sourcing analysis in the supply chain with consideration of capacity disruptions, big data and demand patterns[J]. International Journal of Integrated Supply Management, 2017, 11(1):24.
  4. Ivanov, D. Disruption tails and revival policies: A simulation analysis of supply chain design and production-ordering systems in the recovery and post-disruption periods[J]. Computers & Industrial Engineering, 2019, 127:558-570. doi:10.1016/j.cie.2018.10.043.

Comment (3):  

The clarity of some figures is missing. Please review all figures and improve their quality, clarity, and readability. For example, the legends of some figures are hard to read.

 Response: Thanks for your question. We have modified it according to your request.

Comment (4):  

An extensive proofreading is required.

Response: Thanks for your question! We have modified it according to your request.

We tried our best to improve the manuscript and made some changes in the manuscript. These changes will not influence the content and framework of the paper. We appreciate for Reviewer #4s’ warm work earnestly, and hope that the correction will meet with approval. Once again, thank you very much for your comments and suggestions.

Yours,

Yi zheng
